# Eco-Toxicological and Kinetic Evaluation of TiO₂ and ZnO Nanophotocatalysts in Degradation of Organic Dye

**Sajjad Khezrianjoo [1,†], Jechan Lee [2,†], Ki-Hyun Kim [1,*] and Vanish Kumar [3,*]**

1   Air Quality & Materials Application Lab, Department of Civil & Environmental Engineering, Hanyang University, 222 Wangsimni-Ro, Seoul 04763, Korea; khezrianjoo@gmail.com
2   Department of Environmental and Safety Engineering, Ajou University, 206 Worldcup-ro, Suwon 16499, Korea; jlee83@ajou.ac.kr
3   National Agri-Food Biotechnology Institute (NABI), S.A.S. Nagar, Punjab 140306, India
*   Correspondence: kkim61@hanyang.ac.kr (K.-H.K.); vanish.saini01@gmail.com (V.K.)
†   These authors are co-first authors as they contributed equally.

**Abstract:** In this study, the photocatalytic degradation of azo dye "Food Black 1" (FB1) was investigated using TiO₂ and ZnO nanoparticles under ultraviolet (UV) light. The performances of the two photocatalysts were evaluated in terms of key parameters (e.g., decolorization, dearomatization, mineralization, and detoxification of dye) in relation to variables including pre-adsorption period, pH, and temperature. Under acidic conditions (pH 5), the ZnO catalyst underwent photocorrosion to increase the concentration of zinc ions in the system, thereby increasing the toxic properties of the treated effluent. In contrast, TiO₂ efficiently catalyzed the degradation of the dye at pH 5 following the Langmuir–Hinshelwood (L–H) kinetic model. The overall results of this study indicate that the decolorization rate of TiO₂ on the target dye was far superior to ZnO (i.e., by 1.5 times) at optimum catalyst loading under UV light.

**Keywords:** azo food dye; eco-toxicity; Langmuir–Hinshelwood kinetics; photocatalysis; photocorrosion; nanoparticles

## 1. Introduction

It is well-known that diverse forms of synthetic dyes are used in food industries to add colors to various products, including sports drinks, candies, and baked eatables [1]. Food dyes and pigments amounting to approximately $8 \times 10^5$ t are produced annually [1]. Meanwhile, daily consumption of artificial food dyes per capita has increased by 500% over the past 50 years according to the Food and Drug Administration (FDA) [2]. Food dyes have no nutrients, and they contain aromatic rings and azo groups [1,3]. Some food dyes are mutagenic, carcinogenic, and genotoxic [1,4]. Among different food dyes, Food Black 1 (FB1) azo dye is known to have a cytotoxic effect on cells at high concentrations (i.e., 6.67 mM) [5]. This dye was found to have harmful effects even at lower concentrations (e.g., the level of estradiol decreases at 1 mM of FB1) [5]. In light of such undesirable effects, it has become crucial to treat the effluents containing food dyes discharged from food industry facilities.

Several technological options (e.g., biological, physical, and chemical) have been employed for the removal of noxious dyes in wastewater [6–8]. Among physical methods, adsorption on carbonaceous materials (e.g., carbon nanotubes and activated carbon) was considered as an efficient dye removal technique [6,7], while enzyme degradation was similarly considered efficient as a biological technique [8]. Among the chemical techniques, heterogeneous photocatalytic oxidation (HPO), such as the method developed based on TiO₂ and ZnO photocatalysts, has been proposed as an efficient

option to treat various organic dyes under UV light [9–12]. A general mechanism for HPO of azo dyes under the UV illumination using ZnO catalyst is depicted in Figure 1 [13].

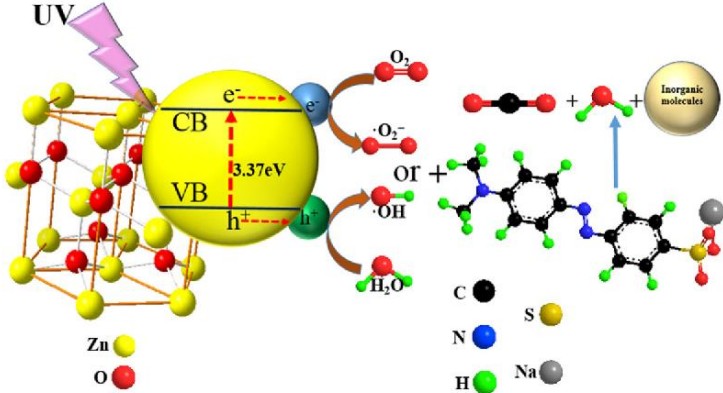

**Figure 1.** General mechanism for photocatalytic degradation of azo dyes under UV illumination on ZnO catalyst. Reproduced from [13] with permission from Springer.

In HPO, the photocatalyst absorbs light energy to excite electrons to the conductive band. This process generates an electron–hole pair, which could react with water, hydroxyl groups, and oxygen molecules to generate highly reactive oxygen species such as hydroxyl radicals and superoxide anions. Thereafter, these radical species attack vital organic components (e.g., dyes) to decompose them through oxidation reactions [9–13]. In light of the energy-intensive nature of HPO, its combination with biological processes (e.g., anaerobic, aerobic, or their combination) can considerably reduce the operational cost due to the low energy consumption of the biological processes [14]. Accordingly, the detection of their destructive intermediates and the evolution of toxicity over the course of the HPO process become important. Modern analytical techniques (e.g., LC-UV/Vis, RP-HPLC, GC-MS, and LC-MS) can be used for such applications. The toxicological evaluation is essential to properly regulate the quality of the discharged wastewater. In light of such demand, methods have been developed for estimating the toxicity of discharged wastewater [15]. Seed germination and root growth of the *Lepidium sativum* L. (*L. sativum* L.) plant was considered in this work as a direct method for evaluating the toxicity of dye pollutants and their degradation byproducts [16,17]; this plant is known for its many advantageous properties for such tests (e.g., high growth rate and ability to grow in a broad pH range from 3.5 to 10) [18].

To date, enormous efforts have been made to compare photocatalytic performance in HPOs of dyes between diverse photocatalysts like $TiO_2$ and ZnO [19,20]. However, relatively little is known about various issues related to such processes (e.g., the generated intermediates of the dye-treated samples, evolution of their toxic properties, and their biodegradability) [21]. To help gain a better knowledge of such processes, we studied the detoxification of FB1 azo dye by photocatalytic degradation over two different ($TiO_2$ and ZnO) nanoparticles. The performance of each of the two photocatalysts was assessed in terms of decolorization, mineralization, and detoxification for HPO based on evaluations of the activation energy, detoxification efficiency, and reaction kinetics. Moreover, the influence of key variables (e.g., pre-adsorption time, initial pH of dye solution, and temperature) on the degradation/detoxification processes was studied to gain a better understanding of their HPO processes. Furthermore, the effects of adsorption on both applied catalysts have also been assessed under different pH conditions because HPO is a surface phenomenon. It was previously reported that perfect decolorization and a considerable decrease in chemical oxygen demand (COD) in the dye solution could be achieved using both $TiO_2$ and ZnO catalysts and optimizing the operating conditions [22]. It was also observed that the toxicity of the treated solution increased during photodegradation after pre-adsorption of dye on both catalysts. In the case of the UV-ZnO process in acidic medium (pH 5), solution toxicity increased substantially with decolorization of the dye.

Thus, the pH value of the suspension was adjusted before illumination to assess the optimum pH value for the detoxification process. We investigated the comparative performance of two well-known photocatalysts, $TiO_2$ and ZnO, to learn more about diverse options to overcome the limitations of photocatalytic applications. As such, the results of this study are expected to offer valuable insights into the effects of controlling variables on photocatalytic efficiency optimization in the treatment of common pollutants in water systems.

## 2. Results and Discussion

### 2.1. Effect of Catalyst Loading

The change in rate constant ($k_{obs}$) for photocatalytic degradation of FB1 (50 mg/L) is shown for both $TiO_2$ and ZnO catalysts as a function of catalyst dosage in Figure 2a. Accordingly, the degradation rate increased steeply with an increase in $TiO_2$ concentration up to 0.8 g/L. In contrast, the degradation rate increased modestly with an increase in concentration of ZnO to 1.2 g/L. At pH 6.7, the optimum catalyst loadings were 0.8 g/L $TiO_2$ and 1.2 g/L ZnO, which provided approximately 96% and 95% of FB1 degradation in irradiation times of 180 and 240 min, respectively. The dye degradation efficiency (without any catalyst) was low (5% decolorization after 240 min of illumination). Increasing the catalyst loadings to their optimum levels (0.8 and 1.2 g/L for $TiO_2$ and ZnO, respectively) enhanced the reaction rate with an increase in active sites for the catalyst and the generation of more hydroxyl radicals. Meanwhile, the adsorption of dye molecules on the catalyst surface increased. An additional loading of catalyst might increase the opacity of the suspension, thereby decreasing the photocatalytic rate [10,23].

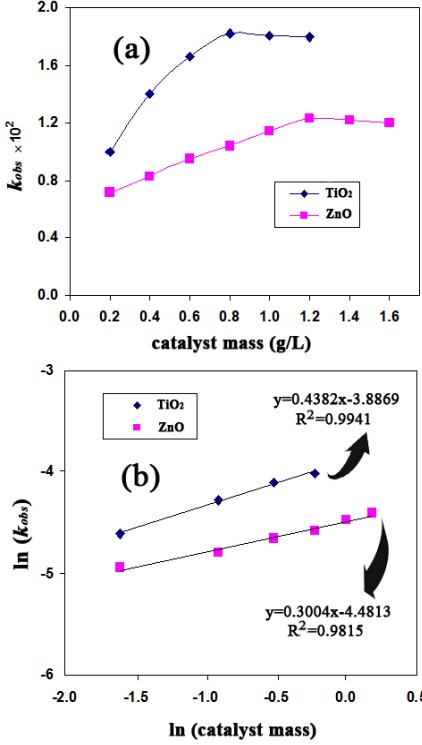

**Figure 2.** Effects of the amount of catalyst ($TiO_2$ or ZnO) on (**a**) heterogeneous photocatalytic oxidation (HPO) rate constant and (**b**) relationship between ln($k_{obs}$) and ln(catalyst mass); $[FB1]_0$ = 50 mg/L, T = 25 °C and pH = 6.7. FB1, Food Black 1.

The relationship between the initial decolorization rate and catalyst loading can be described as ($r_0 \alpha [catalyst]^n$) at catalyst concentrations lower than or equal to the optimum concentration [23]. As seen in Figure 2b, this relationship could be described as $r_0 \alpha [TiO_2]^{0.44}$ and $r_0 \alpha [ZnO]^{0.30}$ for the

UV-TiO$_2$ and UV-ZnO processes, respectively. Therefore, the decolorization of FB1 is more dependent on the catalyst concentration in the UV-TiO$_2$ system than in the UV-ZnO system.

## 2.2. Effect of Pre-Adsorption Time

The effects of pre-adsorption time on decolorization, mineralization, dearomatization, and detoxification of FB1 were evaluated using TiO$_2$ and ZnO catalysts at ambient temperature and pH 6.7. As seen in Figure 3a, the mineralization process was monitored based on COD values. Additionally, dearomatization processes were monitored by measuring the UV peak intensity at around 300 nm (Figure 3b). The toxic properties of FB1 and its degradation byproducts were assessed based on *GI*% (Figure 3c). The initial concentration of FB1 was 50 mg/L, while the initial COD and *GI*% values for this solution were 56 mg O$_2$/L and 10%, respectively.

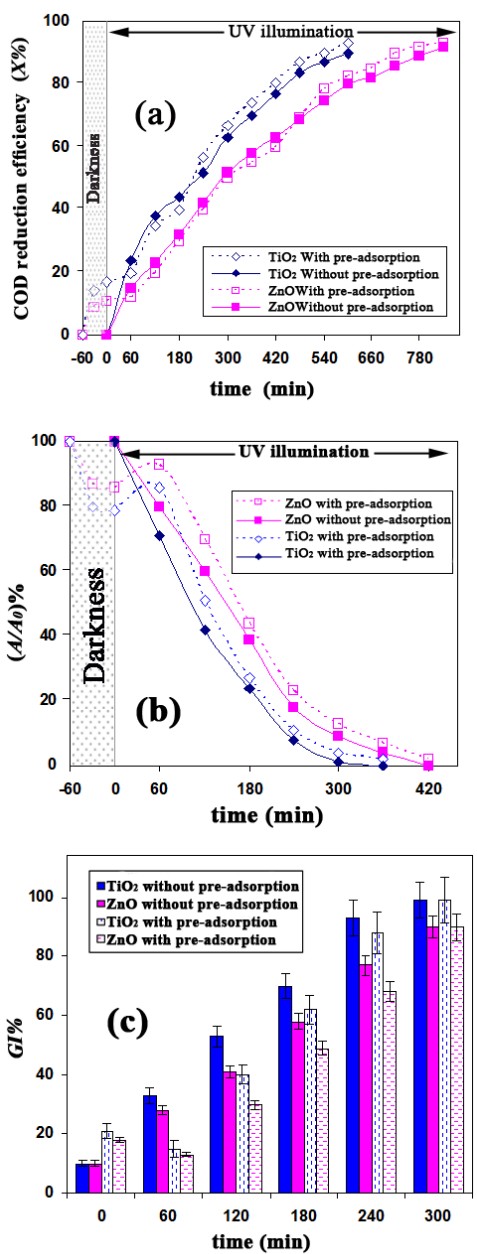

**Figure 3.** Effects of pre-adsorption period on (**a**) mineralization; (**b**) dearomatization; and (**c**) detoxification of FB1 by UV-TiO$_2$ and UV-ZnO processes; [FB1]$_0$ = 50 mg/L, [TiO$_2$] = 0.8 g/L, [ZnO] = 1.2 g/L, T = 25 °C and pH = 6.7.

Using both catalysts (with and without pre-adsorption), almost complete disappearance (more than 95% reduction) of the absorbance peak (at 570 nm) was observed at the end of the run (after 240 min of illumination). This could be due to degradation of the main chromophores (nitrogen–nitrogen double bond N=N). The UV-Vis spectra and images of germination/root growth of *L. sativum* L. during the photocatalytic degradation of FB1 are provided in Figures S1 and S2, respectively.

In the experiments without a pre-adsorption period, a known amount of either $TiO_2$ (0.8 g/L) or ZnO (1.2 g/L) was added to the FB1 solution (initial dye concentration of 50 mg/L), and the lamp was immediately switched on. In these experiments, COD value, peak intensity at 300 nm ($UV_{300}$), and toxic properties of the treated solution decreased continuously with increasing concentrations of $TiO_2$ and ZnO catalysts. As shown in Figure S3, the rate of COD reduction analyzed using the pseudo-first-order kinetic model yielded the rate constants of $3.7 \times 10^{-3}$ $min^{-1}$ for $TiO_2$ and $2.6 \times 10^{-3}$ $min^{-1}$ for ZnO. It is known that the azo linkage (–N=N–) is particularly susceptible to electrophilic attack by hydroxyl radicals [24]. Thus, the formation of aromatic byproducts (e.g., substituted phenols and aromatic hydroxyl amines) is expected as a result of azo band cleavage due to photocatalytic degradation of the azo dye [25]. However, the absorbance in the UV region (about 300 nm) did not increase during the decolorization of FB1 for experiments performed without a pre-adsorption period (Figure 3b). In line with a previous study of the photocatalytic degradation of azo dyes, the vast number of intermediate compounds (as identified by the LC-MS technique) had high electron density aromatic rings or they were negatively charged [26]. Accordingly, a wide range of generated intermediates in HPO of FB1 could also have a negative charge. Thus, formatted intermediates could be absorbed on a $TiO_2$ surface that is positively charged at a pH of 6.7. Similar results (no increase in UV region absorbance) were also observed in a previous study during the UV-$TiO_2$ decolorization of azo dyes such as Orange II, Methyl Red Sodium Salt, Acid Red 183, and Biebrich Scarlet [24]. Meanwhile, in UV-$TiO_2$-based processes, the disappearance of an absorption peak at 300 nm was faster than that in a UV-ZnO-based process. Therefore, degradation of the generated by products that contain benzoic and naphthalene-type rings proceeded faster in the presence of $TiO_2$. However, the possibility of their absorption on the $TiO_2$ surface cannot be excluded. Figure 3b,c indicates that the dearomatization and detoxification trends are in reasonably good agreement during the HPO of FB1.

In the experiments conducted without a pre-adsorption period, the toxicity of the solution as well as the $UV_{300}$ peak intensity decreased continuously. However, the toxicity of the solution and $UV_{300}$ peak intensity increased slightly during the first 60 min of illumination in experiments after the pre-adsorption period, although such trends started to decrease after that time. In general, the trend for $UV_{300}$ removal was nearly identical to that of the toxicity parameter. Complete detoxification was also obtained whenever $UV_{300}$ abatement was high. As reported in a related study, the toxicity of Alachlor ($C_{14}H_{20}ClNO_2$) decreased steeply after its complete dearomatization [27]. Because the $UV_{300}$ is closely related to the toxicity reduction, it could be used as the control parameter for detoxification of the tested compound (i.e., FB1). Similar observations were obtained for $TiO_2$-based studies on other aromatic dyes, e.g., acid blue 25 [28,29]. Figure 3 demonstrates that pre-adsorption time caused a considerable difference in efficiency of mineralization, dearomatization, and detoxification of FB1 during the first 60 min of illumination in both $TiO_2$ and ZnO catalysts. However, no tangible differences were observed in COD $UV_{300}$ or the *GI* parameters for longer illumination times (more than 120 min), irrespective of the pre-adsorption period.

The active sites of the catalyst surface were occupied by the adsorbed dye molecules in the first minutes of illumination. Moreover, slow diffusion of the intermediates generated from the surface of the catalyst can further facilitate the deactivation of active sites of the catalyst. Meanwhile, desorption of dye molecules from catalyst nanoparticles would occur simultaneously during heterogeneous degradation [30]. The reaction would take place on the catalyst surface in a bulk solution of the mixed dye [31]. Therefore, generated intermediates should have less chance to be adsorbed on the catalyst surface and will rather remain in the bulk solution.

## 2.3. Effect of pH

The pH plays an important role in determining both the characteristics of azo dye wastes and the amount of hydroxyl radicals generated during the HPO [31]. The point of zero charge (*Pzc*) in water is around a pH of 6.8 for $TiO_2$ and is around pH 9 for ZnO [10,32]. Therefore, the particle surface is positively charged at pH < *Pzc*, while it is negatively charged at pH > *Pzc*. The effects of pH on the decolorization and mineralization rate constants and dye adsorption on the catalyst surface were determined, as shown in Table 1.

**Table 1.** Comparison of photocatalytic performance of two photocatalysts: $TiO_2$ and ZnO ([FB1]$_0$ = 50 mg/L, [$TiO_2$] = 0.8 g/L, [ZnO] = 1.2 g/L; and T = 25 °C). Detoxification and dearomatization of FB1 in various irradiation times, as well as decolorization/mineralization and dark surface adsorption of FB1 at two different pH values of 5 and 8.

| Irradiation Time (min) | GI% in pH of 5 for | | GI% in pH of 8 for | | (A/A$_0$)% in pH of 5 for | | (A/A$_0$)% in pH of 8 for | |
|---|---|---|---|---|---|---|---|---|
| | TiO$_2$ | ZnO | TiO$_2$ | ZnO | TiO$_2$ | ZnO | TiO$_2$ | ZnO |
| 1 60 | 36 | 20 | 29 | 38 | 68 | 89 | 80 | 71 |
| 2 120 | 57 | 32 | 42 | 60 | 37 | 77 | 51 | 50 |
| 3 180 | 74 | 21 | 59 | 76 | 21 | 65 | 37 | 28 |
| 4 240 | 95 | 22 | 73 | 97 | 5 | 54 | 19 | 6 |
| 5 300 | - | 24 | 92 | - | 0 | 42 | 10 | 0 |
| $k_{dtox}$ (1/min) | 0.345 | - | 0.266 | 0.352 | - | - | - | - |
| Decolorization rate × 10$^2$ (1/min) | 1.46 | 0.73 | 1.97 | 1.56 | - | - | - | - |
| COD reduction rate × 10$^3$ (1/min) | 3.91 | 1.79 | 4.12 | 3.50 | - | - | - | - |
| Dark adsorption % after 60 min | 24.1 | 15.5 | 7.3 | 12.8 | - | - | - | - |

To evaluate surface adsorption, a dye solution with an initial concentration of 50 mg/L was circulated in the reactor for 60 min in darkness. The dark surface adsorption (in percentage) of FB1 on the ZnO surface increased with an increase in the pH of the solution from 5 to 6.7 to 8. The ZnO surface is positively charged in the pH range from 5 to 8. Thus, the electrostatic attraction between negatively charged sulfonic groups and the positively charged ZnO surface led to appropriate adsorption of FB1 on the catalyst surface. After 60 min, almost 14% of the FB1 was adsorbed on the ZnO surface at pH 6.7 (please refer to Table 1 for detailed information). However, there was a great difference between the amount of dark surface adsorption in acidic and alkaline media using $TiO_2$ catalyst (Table 1). At pH 5 and 6.7, the $TiO_2$ surface is positively charged and adsorption is active. Nearly 21% of the dye was adsorbed on the $TiO_2$ surface at pH 6.7. At pH 8, the $TiO_2$ surface becomes negatively charged, exhibiting a low tendency of adsorption for FB1 due to electrostatic repulsion forces with the sulfonic groups of FB1.

Using the UV-$TiO_2$ process at pH 6.7, the decolorization and mineralization rate constants were $1.82 \times 10^{-2}$ and $3.69 \times 10^{-3}$ 1/min, respectively, while those for the UV-ZnO process were $1.23 \times 10^{-2}$ and $2.70 \times 10^{-3}$ 1/min, respectively. The rate of photocatalytic decolorization has been increasing with an increase in pH from 5 to 8 for $TiO_2$ and ZnO (Table 1). Using a UV-$TiO_2$ system-based process, the rate of decolorization in the acidic medium (pH = 5) was lower than that near neutral pH (6.7), although mineralization rate exhibited a reversed pattern. At low pH (acidic medium), the hydrogen ions interacted with an azo linkage to decrease the electron densities at the azo group. As the electrophilic mechanism decreases the reactivity of hydroxyl radicals, it will reduce the azo bond cleavage as well as the decolorization rate [25,33]. The higher mineralization rate of $3.91 \times 10^{-3}$ 1/min at pH 5 could be accounted for by the improvements in the formation of hydroxyl radicals in an acidic medium. The degradation of intermediates via positive holes could also be facilitated due to the positive charges on the $TiO_2$ surface [10].

The rates of decolorization and mineralization over ZnO decreased dramatically at pH 5. This drop is suspected to reflect the effect of photocorrosion of ZnO particles, which is the main drawback of

using ZnO as a photocatalyst [34]. In the case of the UV-TiO$_2$ system, the efficiencies of decolorization and mineralization at pH 8 increased relative to the efficiencies at pH values of 5 and 6.7. The same was observed for the UV-ZnO-based process. More hydroxyl radicals could be generated under an alkaline environment (pH = 8) by oxidizing the available hydroxide ions on the catalyst surface. Therefore, the efficiency of the HPO process can be upgraded on a logical basis [23,25]. The rates of decolorization and mineralization over TiO$_2$ at pH 8 increased by 8% and 11% relative to neutral pH (6.7), while their counterparts for ZnO exhibited enhancements of 26% and 34%, respectively. These overall results indicate that the photocatalytic efficiency of ZnO is more sensitive to the pH of the solution than that of TiO$_2$ regardless of pH.

The effect of initial pH on the detoxification and dearomatization of FB1 was evaluated between the pH values of 5 and 8 (Table 1). The rate of ZnO-based detoxification followed a zero-order kinetic model except for at a pH of 5 (Figure S4). The slope of the linear variations is equal to the $k_{detox}$. Therefore, *GI*% at any time can be determined by Equation (1).

$$[GI\%] = k_{detox}t + [GI\%] \tag{1}$$

Here, [*GI*] and [*GI*]$_0$ are the initial and germination index at a given time, and $k_{dtox}$ is the detoxification rate constant. $k_{detox}$ was calculated to be 0.338 and 0.270 1/min at pH 6.7 for TiO$_2$ and ZnO catalysts, respectively. The detoxification rate constants for both TiO$_2$ and ZnO photocatalysts tested in both acidic (pH 5) and alkaline media (pH 8) are presented in Table 1.

As mentioned in Section 2.2, the intermediates formed during HPO of FB1 may have negative charges. Hence, the adsorption of intermediates on the TiO$_2$ surface at a pH of 8 could be reduced, and the generated intermediates would remain in the solution. As shown in Table 1, the detoxification of FB1 by ZnO catalyst in an acidic medium (pH = 5) showed a complicated trend. In the first 120 min of illumination, the toxicity of the solution decreased, although the detoxification rate at pH 5 was lower than at pH 6.7. The toxicity of the solution began to increase after 120 min of irradiation. Nonetheless, the toxicity of the effluent was not changed considerably at illumination times between 180 and 300 min. On the other hand, the COD and UV$_{300}$ peak intensity value decreased continuously according to Figure 3a,b. As mentioned in Section 3.3, high concentrations of zinc ions can be toxic for plants. Therefore, an increase in toxicity could be due to the increase in Zn$^{2+}$ ions in the solution. The COD values of 24 mg O$_2$/L (approximately 43% reduction) and the UV$_{300}$ peak intensity of A/A$_0$ = 43% indicate that some intermediates and aromatic compounds still exist in the medium after 300 min of illumination. Therefore, the toxic zinc-chelating ligands would also be produced in this solution based on an increase in the toxicity of the solution in a period of between 120 and 300 min. It was reported that the release of Zn$^{2+}$ ions and their adhesion to the cell membrane can cause mechanical damage to the cell wall [35].

### 2.4. Kinetic Studies

The effects of various initial concentrations of FB1 on the photocatalytic degradation rate in the presence of TiO$_2$ (0.8 g/L) or ZnO (1.2 g/L) have been investigated at initial pH values of 5, 6.7, and 8. However, kinetic studies for UV-ZnO photocatalytic degradation of FB1 were not done at pH 5 in light of the photocorrosion and associated change in ZnO attributes at a pH of 5. The experimental data have been rationalized in terms of the L–H kinetic model to describe the solid–liquid reaction. The $k_c$ and $K_{LH}$ constants are presented in both Figure S5 and Table 2. These values were obtained using a least square best fitting procedure (initial FB1 concentration: >50 mg/L).

In many studies, the L–H kinetic model has been used to describe the HPO reaction rate with various concentrations of a target substrate [10,36,37]. However, if the kinetic adsorption equilibrium constant is different from that obtained in the dark, the L–H kinetic model cannot be adopted [23]. Therefore, the influence of the different initial concentrations of FB1 on the surface adsorption has been studied at the same pH values (5, 6.7, and 8). The data obtained from the dark adsorption

experiments were fitted to the Langmuir equation at an initial concentration of FB1 lower than 50 mg/L. Consequently, the Langmuir adsorption constant ($K_{ads}$) and maximum absorbable dye quantities ($Q_{max}$) for the examined pH values were determined (Table 2). The adsorption–desorption equilibrium of FB1 over both catalyst surfaces occurred in a relatively short time period. It was reported that the adsorption process could be described by a film diffusion model over short times [38]. The adsorption of FB1 over the $TiO_2$ catalyst surface occurred within approximately 30, 35, and 50 min at initial pH values of 5, 6.7, and 8, respectively. In contrast, these values were about 35, 35, and 40 min over the ZnO catalyst surface at initial pH values of 5, 6.7, and 8, respectively. These observations indicate that dye adsorption equilibrium time decreases with an increase in the $Q_{max}$ when using $TiO_2$ or ZnO catalysts. The results for the dark adsorption of FB1 on the $TiO_2$ and ZnO catalyst surfaces, in the initial concentrations of 10, 20, 30, 40, and 50 mg/L, have been illustrated in Figures S6 and S7. Moreover, more details about calculating the $Q_{max}$ and $K_{ads}$ constants have also been provided in the Supplementary Materials (Figure S8).

**Table 2.** Determination of the L–H equilibrium adsorption constant ($K_{LH}$ in L/mg), the rate constant of the surface reaction ($k_c$ in mg/L min), Langmuir adsorption constant ($K_{ads}$ in L/mg), and maximum absorbable dye quantities ($Q_{max}$ in mg/g) of FB1 at different pH values ([$TiO_2$] = 0.8 g/L, [ZnO] = 1.2 g/L, and T = 25 °C).

| | | $TiO_2$ | | | | ZnO | | | |
|---|---|---|---|---|---|---|---|---|---|
| | Initial pH | $k_c$ | $K_{LH}$ | $Q_{max}$ | $K_{ads}$ | $k_c$ | $K_{LH}$ | $Q_{max}$ | $K_{ads}$ |
| 1 | 5 | 0.908 | 0.533 | 15.74 | 0.506 | - | - | 7.17 | 0.233 |
| 2 | 6.7 | 0.733 | 0.614 | 14.30 | 0.305 | 0.622 | 0.418 | 6.85 | 0.143 |
| 3 | 8 | 1.032 | 0.860 | 5.29 | 0.145 | 0.803 | 0.760 | 6.41 | 0.122 |

The adsorption constants of many organic compounds under HPO were also determined to be very different (much larger or smaller) from the adsorption constant measured in the dark [38–40]. We also made similar observations for the tested nanomaterials on FB1. Using both UV-$TiO_2$ and UV-ZnO processes, the adsorption constants in the kinetic model were much larger than those in the dark. Nonetheless, the reasons for the various adsorption constants at dark and light conditions are still debatable, although some explanations have been provided. For instance, the adsorptive sites on the catalyst surface could change as a result of UV illumination. The number of these sites would be few to start the reaction, so adsorption constant in the kinetic model may be far different from that obtained in the dark. Meanwhile, the reaction takes place both at the surface of catalyst and in the bulk solution [41]. However, using the UV-$TiO_2$ process at pH 5, adsorption constants in the kinetic and dark models were close to each other ($K_{LH}$ = 1.05 $K_{ads}$), and the HPO of FB1 satisfactorily followed the L–H kinetic model.

### 2.5. Durability of the Catalyst Nanoparticles

Cyclic experiments for the photocatalytic degradation of FB1 using both catalysts ($TiO_2$ and ZnO) were conducted at pH values of 5, 6.7, and 8. After each cycle, the catalyst used was separated by centrifuging, washing with deionized water and drying at 105 °C for 8 h. As shown in Figure 4, the photocatalytic efficiency of ZnO was not altered considerably at pH 8 after three cycles, as the photocatalytic efficiency of ZnO nanoparticles was reduced by less than 3% at neutral pH (6.7). However, the ZnO system at pH 5 degraded 77% of the FB1 molecules after 180 min of UV illumination. Thereafter, catalyst particles were separated, washed, and dried. The obtained catalyst was used in the second cycle at the same condition (180 min of irradiation). At an initial concentration of 50 mg/L, the degradation percentages in the second and third cycle were reduced to 39% and 14%, respectively. As explained above, this reduction is likely to reflect the effect of photocorrosion under acidic conditions. In contrast, 92% of FB1 was removed by the UV-$TiO_2$ process under comparable conditions (pH 5 after three cycles). Meanwhile, approximately 96% and 97% of dye was removed after

three cycles using the UV-TiO$_2$ process at pH values of 6.7 and 8, respectively, without a significant change in the photocatalytic performance.

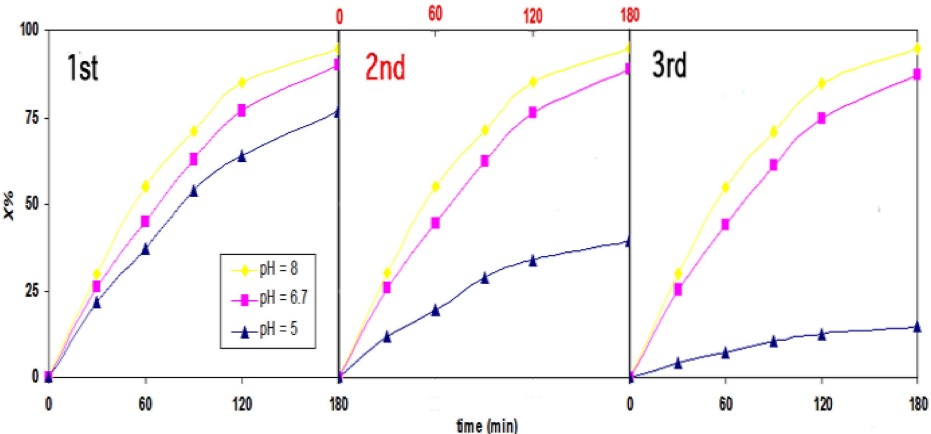

**Figure 4.** The photostability of ZnO nanoparticles based on an investigation of photocatalytic activity under UV illumination at pH values of 5, 6.7 and 8; [FB1]$_0$ = 50 mg/L) [ZnO]$_0$ = 1.2 g/L and T = 25 °C.

*2.6. Solubility of ZnO in Darkness and its Photocorrosion*

The solubility of ZnO nanoparticles was determined in suspensions across varying concentrations of dye (0, 30, and 50 mg/L). The experiments were conducted in the dark and under UVC illumination (254 nm). The dissolution of ZnO in an acidic medium under dark conditions followed the mechanism shown in Equations (2) and (8) [42].

$$ZnO + 2H^+ \rightleftharpoons Zn^{2+} + H_2O \tag{2}$$

$$ZnO + H^+ \rightleftharpoons Zn(OH)^+ \tag{3}$$

The photodecomposition of UV-ZnO could generate Zn$^{2+}$ ions. This reaction occurs in four steps involving dissolution of ZnO (Equation (4)) and positively charged holes ($h^+$) into surface oxygen and zinc ions [11].

$$ZnO + 2h^+ \rightleftharpoons Zn^{2+} + 0.5O_2 \tag{4}$$

Note that the oxidation potential of ZnO does not depend on pH, while its photodecomposition depends on pH (Equation (5)) [42,43]:

$$ZnO + nH_2O + 2h^+ \rightleftharpoons Zn(OH)_2^{(2-n)+} + 0.5O_2 + nH^+, \tag{5}$$

where *n* depends on pH.

The dissolution of ZnO nanoparticles (1.2 g/L) in the dark was observed in circulated aqueous suspensions with pH values of 5 and 6.7 with different dye concentrations (0, 30, and 50 mg/L). The equilibrated concentrations of Zn$^{2+}$ in water at pH values of 6.7 and 5 (after 60 min) were measured as $5.3 \times 10^{-5}$ and $8.9 \times 10^{-5}$ mol/L, respectively. At pH 6.7, the amounts of dissolved Zn$^{2+}$ in suspensions containing 30 and 50 mg/L FB1 were reduced by 14 and 21%, respectively. In an acidic solution (pH = 5), the Zn$^{2+}$ concentrations decreased by 18% and 29% with the addition of 30 and 50 mg/L FB1, respectively, compared to the one without the dye. The equilibrium times of ZnO dissolution in the dark at pH 6.7 in the presence of 30 and 50 mg/L FB1 reached about 70 and 80 min, respectively. However, the equilibrium times for ZnO dissolution in the dark at a pH of 5 in the presence of 30 and 50 mg/L FB1 increased to about 75 and 90 min, respectively. The obtained results indicate that adding the dye molecules to the ZnO suspension can hinder the corrosion process in the dark. The interaction of water with the dye-covered surface could be suppressed because a portion of the ZnO surface can

be covered by the dye molecules. As presented in Table 1, the maximum adsorption of the dye on the ZnO surface in the dark was obtained at a pH of 5, and a maximum reduction in corrosion of ZnO was also observed at this pH.

The effect of the initial dye concentration on the dissolution of ZnO (1.2 g/L) has been studied at pH values of 5 and 6.7 under UVC illumination (254 nm). As shown in Figure 5, a higher concentration of $Zn^{2+}$ was identified when the samples were irradiated over extended times. However, no notable change in $Zn^{2+}$ concentration was observed after reaching equilibrium. In the absence of dye, the $Zn^{2+}$ concentrations at pH values of 5 and 6.7 reached $5.9 \times 10^{-3}$ and $1.7 \times 10^{-4}$ mol/L, respectively, after 120 and 210 min, respectively. The relatively short equilibrium time as well as high concentration of $Zn^{2+}$ at pH = 5 (compared with pH 6.7) also supported the effect of photocorrosion of ZnO. The equilibrium times increased to almost 240 and 300 min in the one liter ZnO suspension (1.2 g/L) at neutral pH (6.7) with the addition of 30 and 50 mg of FB1, respectively. The equilibrium concentrations of $Zn^{2+}$ decreased by 7% and 11% at this pH. Compared to the zinc oxide suspension (1.2 g/L) without a dye compound, almost a 23% reduction in $Zn^{2+}$ concentration took place in the presence of FB1 (30 mg/L) at pH 5, and the equilibrium time was reached in 180 min. However, photocorrosion of ZnO was reduced by 31% at pH 5 for degradation of 50 mg/L of FB1, and the equilibrium time reached 180 min. These observations might be attributed to the adsorption of dye molecules on the catalyst surface, while the reaction of positive holes ($h^+$) with adsorbed dye molecules can suppress the photocorrosion of ZnO. It was reported that the presence of vacant sites on the ZnO surface can facilitate the photocorrosion [42]. The strong adsorption of dye molecules on the ZnO surface likely suppressed its photocorrosion significantly [44]. Photocorrosion of ZnO nanoparticles could be neglected in the absence of water and/or $h^+$ (Equation (5)). In the case of the ZnO photocatalyst, the concentration of $Zn^{2+}$ as well as the toxicity of the solution was not changed significantly at pH 5 after 180 min (Figure 5 and Table 1). Thus, the main factor controlling the effluent toxicity is the concentration of zinc ions.

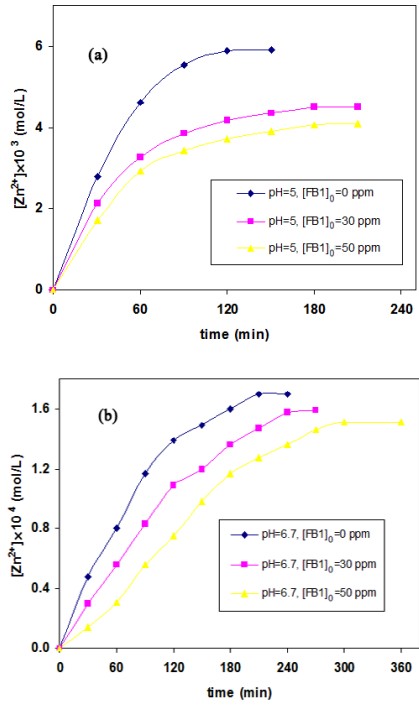

**Figure 5.** Evaluation of the $Zn^{2+}$ concentration produced by photodecomposition of ZnO at two pH values of (**a**) 5 and (**b**) 6.7 across varying FB1 concentration levels under UV irradiation; $[FB1]_0 = 0, 30,$ and 50 mg/L, [ZnO] = 1.2 g/L, and T = 25 °C.

### 2.7. Effect of Temperature

At near-neutral pH (6.7), the effects of dark surface adsorption (and rate constants) on the photocatalytic degradation of FB1 (50 mg/L) were determined using the optimum concentrations of TiO$_2$ (0.8 g/L) or ZnO (1.2 g/L) catalysts at different temperatures (5, 15, 25, 35, and 45 °C), as summarized in Table 3. Operating temperatures higher than 45 °C were not appropriate for these studies due to the possible evaporation of the solution during the experiments. High temperatures reduce the adsorption of FB1 on the surfaces of both catalysts. The suppressed adsorption of FB1 should decrease its photocatalytic oxidation on the surface of catalyst particles. Meanwhile, an increase in temperature decreases the solubility of oxygen in water, which facilitates the electron–hole recombination and is undesirable [36]. However, using both catalysts, the photocatalytic degradation rate of FB1 increased with increasing temperature.

**Table 3.** Effects of temperature on photocatalytic decolorization and mineralization rate as well as dark surface adsorption of FB1; [FB1]$_0$ = 50 mg/L, [TiO$_2$] = 0.8 g/L, [ZnO] = 1.2 g/L and pH = 6.7.

| | Temperature °C | 5 | | 15 | | 35 | | 45 | |
|---|---|---|---|---|---|---|---|---|---|
| | Catalyst Type | TiO$_2$ | ZnO | TiO$_2$ | ZnO | TiO$_2$ | ZnO | TiO$_2$ | ZnO |
| 1 | Dark surface adsorption % | 22.7 | 16.1 | 22.0 | 14.9 | 19.2 | 13.1 | 18.3 | 12.2 |
| 2 | $k_{obs} \times 10^2$ (1/min) | 1.72 | 1.18 | 1.77 | 1.21 | 1.86 | 1.25 | 1.92 | 1.27 |

The apparent activation energy ($E_a$) for the TiO$_2$- and ZnO-based photocatalytic degradation of FB1 has been calculated from the Arrhenius equation (Figure 6); the $E_a$ values were 1.98 and 1.32 kJ/mol, respectively. These results indicate that the former is slightly more sensitive to temperature than the latter.

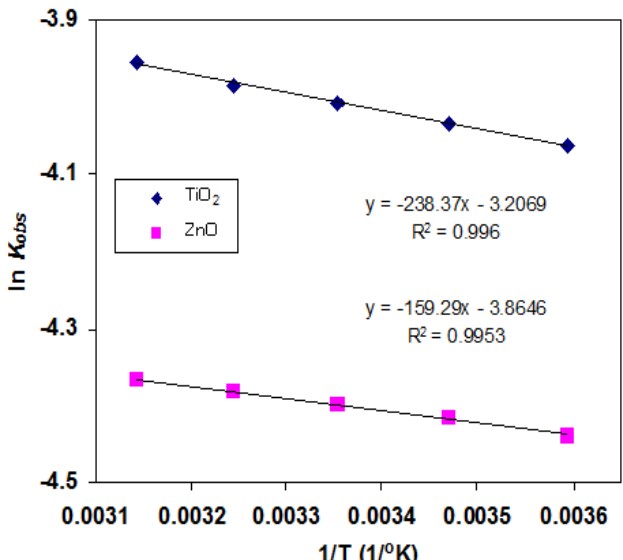

**Figure 6.** Plot of ln($k_{obs}$) versus 1/T for the decolorization and mineralization of FB1 by the HPO process ([FB1]$_0$ = 50 mg/L, [TiO$_2$] = 0.8 g/L, [ZnO] = 1.2 g/L, and pH = 6.7).

To investigate the influence of reaction temperature on detoxification of FB1, the *GI* values for both catalysts were measured at temperatures of 5, 25, and 45 °C. Using samples taken at 1-h intervals, a statistical analysis on the toxicity levels shows that the detoxification efficiency of FB1 by both catalysts was not affected in the temperature range of 5 to 45 °C at the 5% level of significance.

## 2.8. Evaluation of $EC_{50}$ and Biodegradability Tendency

Toxicity effects of pollutants are commonly assessed as a concentration level causing a specific effect (such as death or growth inhabitation) in 50% of the tested organisms (i.e., effective concentration, $EC_{50}$). To obtain this eco-toxicological indicator ($EC_{50}$), the effects of FB1 concentrations (10, 20, 30, 40, and 50 mg/L) on inhibiting *L. sativum* L. root growth (Figure 7a) and seed germination percentage (Figure 7b) were studied. Then, the $EC_{50}$ was calculated according to the method proposed previously [45]. The images of seed germination and root growth of *L. sativum* L. in different concentrations of FB1 are presented in Figure S10. The concentration of FB1 for $EC_{50}$ was determined to be 29.71 ± 0.87 mg/L. As seen in Figure 7c, increase in FB1 concentration caused a decrease in plant height. These observations indicate that FB1 molecules have a toxic effect in the meristematic and parenchyma cells of *L. sativum* L. root and shoot.

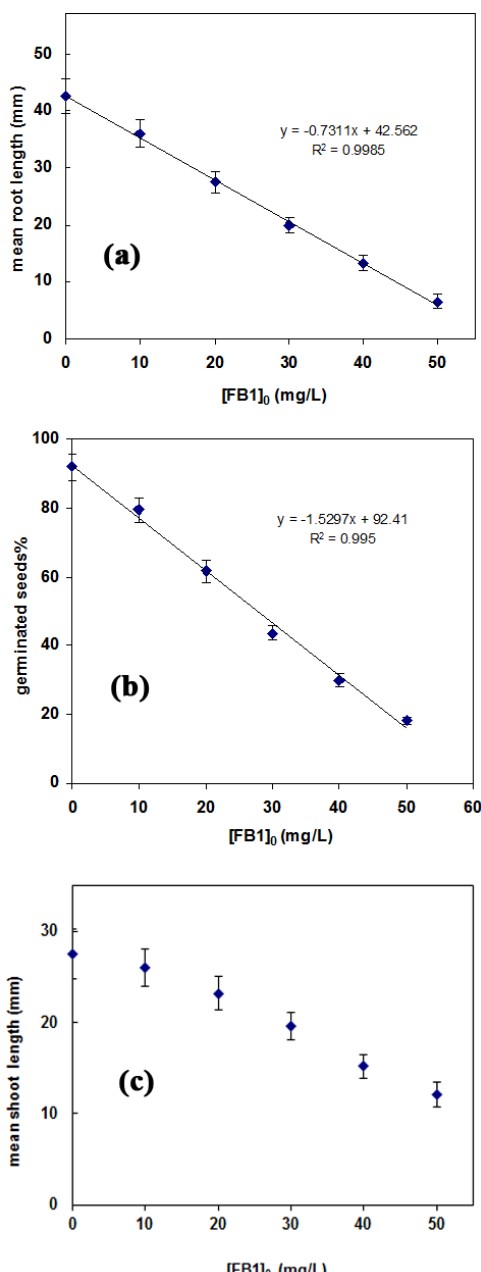

**Figure 7.** Influence of FB1 concentration on (**a**) root length; (**b**) germination; and (**c**) shoot length of *L. sativum* L.; T = 25 °C and pH = 6.7.

The biodegradability of the treated effluent of FB1 at ambient temperature and neutral pH was evaluated based on the ratio of $BOD_5/COD$. It should be noted that the $BOD_5/COD$ ratio is considered to be the biodegradability index (BI) of wastewater [46]. Wastewater is considered biodegradable if the BI value is >0.4. Wastewater is partially biodegradable for BI values between 0.3 and 0.4, while it is non-biodegradable for BI < 0.3 [46]. In the current study, the photocatalytic degradation of FB1 was conducted without a pre-adsorption period with an initial dye concentration of 50 mg/L in the presence of $TiO_2$ (0.8 g/L) or ZnO (1.2 g/L). As discussed in Section 2.2, the GI and BI values for FB1 solution before HPO were 0.10% and 0.13, respectively. This indicates that FB1 with an initial concentration of 50 mg/L is not biodegradable. The GI% in $TiO_2$ (120 min irradiation) and ZnO catalysts (180 min irradiation) reached about 53% and 58%, respectively, implying that the treated solution is toxic. The respective BI values of 0.43 and 0.50 for the two catalysts suggest that the effluent is biodegradable. The performance of these catalysts was also assessed for an extended duration. In the case of $TiO_2$, GI reached 70% after about 180 min of irradiation, implying low toxicity with a corresponding BI of 0.74. In contrast, the GI and BI values in the UV-ZnO process after about 240 min of irradiation were 77% and 0.82, respectively.

## 3. Materials and Methods

### 3.1. Chemicals

Commercially available P25 titanium dioxide ($TiO_2$) was purchased from Degussa Co., Essen, North Rhine-Westphalia, Germany. This material had a BET surface area of 50 $m^2$/g and an average particle diameter of 30 nm. Zinc oxide (ZnO) was purchased from Reinste Company (New Delhi, India); it had a surface area of $30 \pm 5$ $m^2$/g and an average particle diameter of 14 nm. These catalysts were used for experiments without further modification. The FB1 azo dye $C_{28}H_{17}N_5Na_4O_{14}S_4$ (CAS No. 2519-30-4; CI No. 28440; >96% purity) was purchased from Sigma-Aldrich (Bangalore, India) and was used as received. The molecular structure of the dye is shown in Figure S11. The pH of the system was controlled by adding hydrochloric acid (HCl) or sodium hydroxide (NaOH) (Merck product).

### 3.2. Reactor Setup and Photocatalytic Degradation Procedure

A rectangular circulating labmade photo-reactor made of glass (dimensions: 270 mm length, 200 mm width, and 50 mm height) was used. The simulated wastewater solution was prepared by dissolving a known amount of FB1 (in most cases 50 mg) in one liter of distilled water; the initial depth in the solution in reactor was 19 mm. A total of six UV lamps (18 W each) (Philips TUV PL-L, Bangalore, India) were positioned on top of the reactor 7 cm from the surface of the solution. The maximum lamp emission at 254 nm was measured using a TOPCON UV-R-1 spectroradiometer (TOPCON, Tokyo, Japan). All the experiments were operated in a batch mode. The reactor was equipped with a water-flow jacket for regulating the temperature of the solution by means of an external circulating flow. A thermostat bath with an accuracy of about ±0.1 °C was used for adjusting the external circulating water flow temperature. The whole reactor body and internal wall of the reactor case were covered with reflectors made of thin layer polished aluminum. To obtain well-mixed and fluidized catalyst particles, a pump with a flow-rate of about 4300 mL/min was circulating the solution from one corner to the opposite corner during the experiments.

The HPO process cannot be sustained without a constant supply of dissolved oxygen [23]. Thus, air at constant flow-rate was supplied to the four corners of the reactor using a micro air compressor. The schematic of the photo-reactor is shown in Figure S12. To perform an HPO experiment, one liter of a suspension containing 50 mg/L of FB1 and a known amount of catalyst ($TiO_2$ or ZnO) was prepared. The solution was transferred to the reactor to begin degradation. Then, the lamp was switched on while adjusting the temperature and acidity (i.e., pH).

It is well known that adsorption and diffusion have important roles in photocatalytic degradation processes [47]. Moreover, previous studies indicated that photocatalytic degradation processes are

initiated by the adsorption of molecules on the catalyst surface; consequently, a single layer of molecules could be formed on the catalyst surface [47]. The electrostatic interactions that arise from the attraction of charge clouds between FB1 molecules and $TiO_2$ or ZnO surfaces could lead to adsorption of dye on the catalyst surface. To equilibrate the adsorption/desorption process prior to the UV light illumination, the pristine forms of either $TiO_2$ (0.8 g/L) or ZnO (1.2 g/L) nanoparticles were dispersed separately in 50 mg/L solutions of FB1 with circulation of the suspension for 1 h under dark conditions (i.e., no UV light). After one hour there was no photodegradation of the adsorbed FB1 molecules. The FB1-adsorbed molecules were subsequently harvested by centrifugation (2000 rpm for 30 min), and then the concentration of FB1 in the obtained solution was calculated using a UV spectrophotometer and appropriate calibration curves. After circulation, the light was switched on for decolorization and mineralization experiments. Samples (4 mL) were taken at intervals of 15 and 30 min to evaluate decolorization and mineralization of the target dye. Catalyst particles were separated by centrifuging after each sample was taken. Decolorization analyses were performed with a UV-Vis spectrophotometer (Systronics 168, Delhi, India). The absorbance of FB1 was measured at the appropriate maximum wavelength using 1 cm matched quartz cells. The concentration of FB1 was measured based on the Beer–Lambert law. As the change in $\lambda_{max}$ (i.e., 570 nm) was negligible across varying pH values, experiments were carried out at this wavelength. The absorbance peak was nonetheless seen to increase with an increase in pH. The calibration curves for the peak are given in Figure S13.

Chemical oxygen demand (COD) was measured using a closed reflux digester reactor (HACH, DRB 200, CO, USA). Thereafter, the decolorization and mineralization efficiency or conversion (*X*%) of FB1 at any time was obtained by the equation:

$$X\% = \frac{C_0 - C}{C_0} \times 100$$

For the computation of decolorization, the concentrations (mg/L) of FB1 at the beginning and at a given time were $C_0$ and $C$, respectively. Likewise, $C_0$ and $C$ were used as the respective COD values (mg $O_2$/L) in the dye solution to assess the mineralization efficiency. Dearomatization was monitored by measuring the peak intensity of the FB1 solution at a wavelength of around 300 nm, the absorbance peak around this wavelength is attributed to the aromatic rings [48]; here, $A_0$ and $A$ are the peak intensities of the treated solution at the beginning and a given time, respectively.

### 3.3. Eco-Toxicological Evaluations

Eco-toxicological tests were conducted using a standard protocol (ISO/DIS 15 799, (1999)) [16]. The dicotyledonous garden cress *L. sativum* L. was used as an indicator of eco-toxicological evaluation. The tolerance of the seed germination and root growth of *L. sativum* L. to FB1 were measured to evaluate toxicity. The length of shoots was also measured in different concentrations of dye. The seeds were purchased from a local market. A magnifier (×5) was used to see discolored (or physically damaged) seeds. In this way, abnormal seeds were removed. Toxicological experiments were conducted at various pH values (i.e., acidic (pH 5), neutral (pH 6.7), and alkaline (pH 8)) to compare the detoxification process. Simultaneously, a positive control was also performed at the same pH values using sterile distilled water (where no dye compound was added to the mixture).

To determine the extent of germination, the length of the main root was measured with a ruler against a black background. FB1 was maintained at four different concentration levels (10, 20, 30, and 50 mg/L) to obtain the 72-h exposure average effective concentration (72-h $EC_{50}$). Accordingly, a 4 mL solution of a certain concentration of FB1 was added to the containers. To obtain the toxicity of degradation byproducts, solution samples containing 50 mg/L of FB1 were illuminated in the presence of either 0.8 g/L of $TiO_2$ or 1.2 g/L of ZnO. Consequently, samples (4 mL each) were taken at each 60 min interval, and catalyst nanoparticles were removed by centrifugation (2000 rpm for 30 min). The obtained solution was added to the containers. The containers were incubated for 72 h in a dark

growth chamber at ambient temperature and 90% relative humidity. The germination index percentage (*GI*%) was used to record the percentage of germination [49]:

$$GI\% = (\frac{S_S}{S_B})(\frac{L_S}{L_B}) \times 100,$$

where $S_S$ and $S_B$ were the average number of germinated seeds for the sample and blank, respectively, and $L_S$ and $L_B$ were the mean root lengths of seeds for the sample and blank, respectively. A total of 15 seeds were distributed in each Petri dish, and four replicate plates were used for each isolate. All 60 obtained datasets were analyzed using the SPSS (Version 22.0, IBM Corp, Armonk, NY, USA) statistical package software for the determination of *GI*% using the mean root length and average number of seeds germinated. To measure the shoot length, seeds were germinated for 2 days in darkness. Groups of 10 uniform germinated seeds were transferred to the 9-cm Petri dishes and six replicate plates were used for each isolate. Afterwards, 10 mL of distilled water (control) or solution of FB1 (10, 20, 30, 40, and 50 mg/L) was added to the Petri dishes. Seedlings were grown under natural daylight (12 h) for 4 days.

A one-way ANOVA was used to compare the means of two or more independent groups. This method was used to compare the means of different treatments as well as differences in mean values between individual and control samples. As such, the ANOVA provided statistical significance of the obtained data. The stepwise method of multiple regression analysis was also performed using the *F* test with a significance level of 0.05 (*F*-enter = 0.05). It was reported that the *GI*% values of the three classes (i.e., (1) lower than 40%; (2) between 40% and 60%; and (3) higher than 60%) should correspond to high, moderate, and low toxicities, respectively [50,51].

Another important factor that can affect the toxicity of the solution is the concentration of $Zn^{2+}$. Accordingly, the concentration of zinc ions ($Zn^{2+}$) in the solution was also measured using an atomic absorption spectrometer, contrAA 700 (Analytik Jena AG, Jena, Germany). The presence of heavy metal ions such as zinc in water may be beneficial or toxic to the environment depending on its concentration levels [52]. Plants generally require Zn in trace quantities for proper growth. However, free hydrated zinc ion ($Zn^{2+}$) can be toxic to plants at high concentrations [51]. The photocorrosion rate of ZnO particles is associated with the pH of the solution. A decrease in pH increases the ZnO photocorrosion, while no photocorrosion of ZnO takes place at pH ≥10 [18]. Therefore, the concentration of zinc ions (an indicator of ZnO photocorrosion) could be controlled by controlling the solution pH. The dissolution of zinc would also be related to its intrinsic and physicochemical properties (e.g., surface area, particle size, and chemical composition) as well as the environmental parameters of the exposure matrix, such as organic matter and temperature [53]. Thus, a five-day biochemical oxygen demand assay ($BOD_5$) was also carried out in the UV-ZnO process at a pH of 5 to evaluate changes in the biodegradability of samples according to the standard respirometric method after separation of catalyst nanoparticles by centrifuging [54]. The temperature of the experimental system was controlled using Velp Scientifica open circulating baths. PolySeed® inoculum (Interlab®) was used as a blend of broad spectrum bacteria that was designed specifically as a seed inoculum for the $BOD_5$ test. $BOD_5$ was determined for FB1 both from an initial concentration of 50 mg/L and from the degradation byproducts.

### 3.4. Photocatalytic Oxidation Kinetics and Dark Surface Adsorption

The FB1 concentration changed over time as the photocatalytic reaction proceeds. In the meantime, catalyst concentration remained more or less constant. Accordingly, a pseudo-first-order reaction was assumed and applied to explain the reaction rate equation and to determine the reaction rate constants. The photocatalytic degradation of FB1 using both applied catalysts agreed well with the well-known pseudo-first-order kinetic model at pH values of 5, 6.7, and 8 for all the experiments (the obtained rate

constants in Table S1). The L–H kinetics model was used to describe the dependence of the observed reaction rate on the initial FB1 concentrations:

$$\frac{1}{k_{obs}} = \frac{1}{k_c K_{LH}} + \frac{[FB1]_0}{k_c}.$$

Here, $k_{obs}$ is a pseudo-first-order kinetic rate constant, $[FB1]_0$ is the initial concentration of dye (10, 20, 30, 40, and 50 mg/L), $K_{LH}$ is the L–H equilibrium adsorption constant (L/mg), and $k_c$ is the rate constant of the surface reaction (mg/L min).

Surface adsorption is important in evaluating the HPO rate using the L–H kinetic model [23]. At pH values of 5, 6.7, and 8, one liter of a suspension containing the known concentrations of FB1 (10–50 mg/L) was circulated in the presence of $TiO_2$ (0.8 g) or ZnO (1.2 g) in the reactor while the UV light was off (under darkness). Samples were taken every 5 min to find the adsorption–desorption equilibrium time. Then, catalyst particles were removed by centrifuging at 2000 rpm for 30 min. Thereafter, samples were taken after equilibration (60 min of circulation in the darkness). Data obtained from the adsorption experiments were fitted to the modified empirical Langmuir equation:

$$\frac{C}{Q} = \frac{1}{Q_{max} K_{ads}} + \frac{C}{Q_{max}},$$

where $C$ is the equilibrium concentration, $Q_{max}$ is the maximum absorbable dye quantity, and $K_{ads}$ is the equilibrium constant for adsorption (L/mg). The adsorbed quantity ($Q$, mg/g) was calculated as:

$$Q = \frac{V \Delta C}{m},$$

where $\Delta C$ is the difference between the initial ($C_0$) and equilibrium concentration ($C$), $V$ is the solution volume (L), and $m$ is the catalyst loading (g).

## 4. Conclusions

The performances of two commercially available nanophotocatalysts (i.e., P25 $TiO_2$ and ZnO) were evaluated for the heterogeneous photocatalytic degradation of food azo dye "FB1" in a suspended batch reactor. The toxicities of FB1 dye and its photodegradation byproducts were assessed against the *L. sativum* L. garden cress. The effective concentration ($EC_{50}$) of FB1 was determined to be 29.71 ± 0.87 mg/L. When using a pre-adsorption period under dark conditions, there were increases in the toxicity of the solution and $UV_{300}$ peak intensity within the first 1 h of photocatalytic degradation. However, without pre-adsorption, the COD, $UV_{300}$, and toxicity decreased continuously over $TiO_2$ and ZnO. Dearomatization of FB1 during the HPO could be considered as a control parameter for its detoxification. In the case of $TiO_2$, a maximum mineralization and decolorization rate was obtained at pH 8. However, ZnO recorded the highest efficiency in toxicity reduction at a pH of 8. In an acidic medium (pH = 5), an increase in toxicity of FB1 was unexpectedly observed due to the photocorrosion of ZnO particles and formation of $Zn^{2+}$ ions. In both acidic and basic conditions, the photocatalytic activity of $TiO_2$ nanoparticles was generally maintained even after three cycles. In contrast, the performance of ZnO nanoparticles remained stable only in a basic medium (pH 8) after three cycles. The photocatalytic degradation and detoxification of FB1 were not temperature-sensitive for either catalyst. The results suggested that bioassay evaluation before reaching complete mineralization (i.e., through monitoring of effluent toxicity) would be a useful metric for HPO by both $TiO_2$ and ZnO photocatalysts. For both catalysts, the obtained $K_{LH}$ was much larger than $K_{ads}$ in all three tested pH values (5, 6.7, and 8). However, the kinetics of HPO were well fit by the L–H model only in the UV-$TiO_2$ system at a pH of 5. Accordingly, it is desirable to assess reusability and long-term stability of the applied photocatalysts (i.e., $TiO_2$ and ZnO) for dye-polluted effluents with respect to operational parameters (e.g., pH and temperature), along with an evaluation of the toxicity of the treated effluent prior to the large-scale implementation of HPO.

**Supplementary Materials:** The following are available online at http://www.mdpi.com/2073-4344/9/10/871/s1, Figure S1: UV-Vis spectra changes of FB1 at different irradiation times, Figure S2: Root growth and seed germination percentage of *L. sativum* L. in FB1 solution during the HPO process, Figure S3: Pseudo-first-order kinetic rate constants for the photocatalytic mineralization of FB1 using $TiO_2$ and ZnO catalysts, Figure S4: The variation of *GI*% versus time for detoxification of FB1 by $UV-TiO_2$ and UV-ZnO processes at different pH values, Figure S5: Variation of reciprocal of constant rate versus different initial concentrations of FB1 at different pH values, Figure S6: Adsorption isotherm of FB1 on $TiO_2$ surface, Figure S7: Adsorption isotherm of FB1 on the ZnO surface, Figure S8: Establishment of Langmuir monolayer adsorption constants for adsorption of FB1 on $TiO_2$ catalyst at different pH values, Figure S9. Establishment of Langmuir monolayer adsorption constants for adsorption of FB1 on ZnO catalyst at different pH values, Figure S10: Evaluation of effective concentration ($EC_{50}$) for FB1, Figure S11: Chemical structure of Food Black 1 (FB1), Figure S12: Schematic view of the photo-reactor, Figure S13: The calibration chart for measuring FB1 concentration at different pH values of 5, 6.7 and 8. Table S1: Pseudo-first-order rate constant values for the different initial concentrations of FB1 at various pH values.

**Author Contributions:** Experimental design, writing original draft, reviewing & editing of paper was performed by S.K. Reviewing & editing of paper was performed by J.L. and V.K. Supervision, final reviewing & editing acquisition performed by the K.-H.K. All authors discussed the results and contributed to the final manuscript.

**Funding:** We acknowledge the support made by the R & D Center for Green Patrol Technologies through the R & D for Global Top Environmental Technologies program funded by the Ministry of the Environment (Grant No: 2018001850001) as well as by a grant from the National Research Foundation of Korea (NRF) funded by the Ministry of Science, ICT & Future Planning (Grant No: 2016R1E1A1A01940995).

**Acknowledgments:** S.K. wishes to thank M. Khanjani, Faculty of Agriculture, University of Bu-Ali Sina, for his assistance in carrying out the toxicity investigations. SK is also grateful to M. Mohammadi for assistance with statistical analysis. VK acknowledges support from the Department of Science and Technology, New Delhi, India in the form of an INSPIRE Faculty Award.

**Conflicts of Interest:** The authors declare no conflict of interest. The funders had no role in the design of the study; in the collection, analyses, or interpretation of data; in the writing of the manuscript, or in the decision to publish the results.

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
