# Peer review of "Eco-Toxicological and Kinetic Evaluation of TiO2 and ZnO Nanophotocatalysts in Degradation of Organic Dye"

_catalysts, doi:10.3390/catal9100871_

Round 1

Reviewer 1 Report

In this research, the authors represented the photocatalytic degradation of FB-1 by ZnO2 and TiO2. Besides this they discussed the impact of dye concentration on seed germination which is really an amazing task as it is related to the agriculturebut the idea has no novelty as this kind of work has been done previously by so many scientists. However I have some comments which authors must account while revising the manuscript:

1. There are several grammatical mistakes throughout the manuscript. Please correct these before submitting the manuscript.

2. The authors discussed about the impact of root length on the germination (Fig. 6). Shoot length also has impact on the germination. Did the authors do any experiment which can show the impact of shoot length on germination? If didn’t please show the impact of shoot length on germination by a simple experiment and explain.

3. The conclusion part is too long. Please simplify this by reducing some words.

4. Please provide a general mechanism for the degradation of dyes by a schematic arrangement.

5. Complete a thorough review of the abbreviations.

6. Some figures resolution is not so good (Fig. 2). Please provide the high resolution image.

Author Response

[1] Reviewer 1

Comments and Suggestions for Authors

In this research, the authors represented the photocatalytic degradation of FB-1 by ZnO2 and TiO2. Besides this they discussed the impact of dye concentration on seed germination which is really an amazing task as it is related to the agriculture but the idea has no novelty as this kind of work has been done previously by so many scientists. However, I have some comments which authors must account while revising the manuscript:

There are several grammatical mistakes throughout the manuscript. Please correct these before submitting the manuscript.

Response: Thanks for the nice comments. The manuscript has been edited by a professional language editing service called E-World company; so, hopefully it now matches the journal standard.

The authors discussed about the impact of root length on the germination (Fig. 6). Shoot length also has impact on the germination. Did the authors do any experiment which can show the impact of shoot length on germination? If didn’t please show the impact of shoot length on germination by a simple experiment and explain.

Response: As per the comment, (i) in the last paragraph of page 13 (line No. 394-396), it was mentioned that an increase in FB1 concentration caused a decrease in plant shoot height; (ii) in the 1st paragraph of page 14 (line No. 399-401), the influence of FB1 concentration on shoot length of L. sativum L was illustrated in Figure 7c; (iii) in the page 16, paragraph 5 (line No. 502-507) the method for measurement of plant shoot length is mentioned. The seeds were germinated for 2 days in darkness. Groups of 10 uniform germinated seeds have been transferred to the 9 cm Petri dishes. Afterwards, 10 mL of distilled water (control) or solution of FB1 (10, 20, 30, 40, and 50 mg/L) was added to the roots. Seedlings were grown under natural daylight (12 h) for 6 days.

The conclusion part is too long. Please simplify this by reducing some words.

Response: According to the reviewer suggestion, the conclusion part is simplified by reducing some words.

Please provide a general mechanism for the degradation of dyes by a schematic arrangement.

Response: According to the reviewer’s suggestion, a schematic diagram for the photocatalytic degradation of azo dyes was added to the 1st paragraph of page 2. The added Figure (Fig. 1) depicts a general mechanism for photocatalytic degradation of azo dyes under the UV illumination using ZnO catalyst.

Complete a thorough review of the abbreviations.

Response: All the abbreviations were rechecked carefully.

Some figures resolution is not so good (Fig. 2). Please provide the high resolution image.

Response: In the last paragraph of page 4 and 1st paragraph of page 5, the high resolution image was provided for the Figure 3. The number of Fig. 2 was changed to Fig. 3 in the revised version.

Reviewer 2 Report

The paper submitted by Khezrianjoo et al., reports a study on the degradation of organic dye "Food Black 1" by using TiO2 or ZnO nanoparticles. The experimental part is well described as well as the results and the discussion of data appropiate. Nevertheless, the introduction part should be revised. The following references may be added in order to better highlight the issues that the authors would address.

Katheresan et al., J. Env. Chem. Eng. 2018, 6, 4676-4697; Zare et al., J. Nanostruct. Chem., 2015, 5, 227-236; Vuono et al., Chin. J. Chem. Eng., 2017, 25, 523-532.

Author Response

[2] Reviewer 2

Comments and Suggestions for Authors

The paper submitted by Khezrianjoo et al., reports a study on the degradation of organic dye "Food Black 1" by using TiO2 or ZnO nanoparticles. The experimental part is well described as well as the results and the discussion of data appropiate. Nevertheless, the introduction part should be revised. The following references may be added in order to better highlight the issues that the authors would address.

Katheresan et al., J. Env. Chem. Eng. 2018, 6, 4676-4697; Zare et al., J. Nanostruct. Chem., 2015, 5, 227-236; Vuono et al., Chin. J. Chem. Eng., 2017, 25, 523-532. 

Response: Thanks for the nice comments. According to the suggestions, the content has been added in the ending paragraph of page 1 (line No. 37-42). The introduction part is revised and the suggested articles were added to the revised manuscript and to its reference list (references 6-8 in the page 19, line No. 605-613).

Reviewer 3 Report

In this work, the degradation kinetics of an azo dye (FB1) on ZnO and TiO2 photocatalysts is presented. The work provides also an eco-toxicological study, following state-of-the-art evaluation protocols.

The authors have investigated the effect of several variables, such as pH, on the degradation kinetics, providing useful and accurate insights on this chemistry. 

The authors provide also interesting considerations on the pH-dependent adsorption/desorption processes. Such mechanisms are extremely important in catalysis on charged surfaces, and more in general for environmental chemistry. For future works, I would suggest to corroborate these considerations through zeta potential and Dynamic Light Scattering measurements, which are powerful tools for investigating adsorption mechanisms on nanoparticles in suspension.

Before publication, I recommend that the authors increase the resolution of the figures. For example, Figure 2a,b,c is impossible to read in the current status. 

The work is conducted in a diligent way. The methodology applied by the authors is sound. The insights provided by the authors would be useful for the catalysis community.

Overall, my final judgement of the paper is absolutely positive and I would recommend this manuscript to be published on "Catalysts".

Author Response

[3] Reviewer 3

Comments and Suggestions for Authors

[1] In this work, the degradation kinetics of an azo dye (FB1) on ZnO and TiO2 photocatalysts is presented. The work provides also an eco-toxicological study, following state-of-the-art evaluation protocols. The authors have investigated the effect of several variables, such as pH, on the degradation kinetics, providing useful and accurate insights on this chemistry. 

Response: Thanks for the nice comments

[2] The authors provide also interesting considerations on the pH-dependent adsorption/desorption processes. Such mechanisms are extremely important in catalysis on charged surfaces, and more in general for environmental chemistry. For future works, I would suggest to corroborate these considerations through zeta potential and Dynamic Light Scattering measurements, which are powerful tools for investigating adsorption mechanisms on nanoparticles in suspension.

Response: In the related future studies, we plan to explore the suggested contents (applying the zeta potential and Dynamic Light Scattering measurements).  

[3] Before publication, I recommend that the authors increase the resolution of the figures. For example, Figure 2a,b,c is impossible to read in the current status.

Response:  As per comment, in the 1st paragraph of page 5, the high resolution image was provided in Figure 3. The number of Fig. 2 was changed to Fig. 3 in the revised version

[4] The work is conducted in a diligent way. The methodology applied by the authors is sound. The insights provided by the authors would be useful for the catalysis community.

Response: Thanks for the nice comment.

[5] Overall, my final judgement of the paper is absolutely positive and I would recommend this manuscript to be published on "Catalysts".

Response: Thanks for the nice comment.

Round 2

Reviewer 2 Report

Manuscript improved, suitable for publication.